# Clinicopathological and Molecular Investigation of Newcastle Disease Outbreaks in Vaccinated and Non-Vaccinated Broiler Chicken Flocks in Nepal

**DOI:** 10.3390/ani14162423

**Published:** 2024-08-21

**Authors:** Subash Regmi, Rajesh Bhatta, Pushkar Pal, Arvindra Shrestha, Tamás Mató, Bikash Puri, Surya Paudel

**Affiliations:** 1Department of Veterinary Pathology and Clinics, Agriculture and Forestry University, Rampur 44200, Nepal; subash.regmi32@gmail.com (S.R.); rbhatta@afu.edu.np (R.B.); bpuri@afu.edu.np (B.P.); 2Department of Veterinary Pathology, Institute of Agriculture and Animal Science, Rampur 44200, Nepal; arvinshrestha5@gmail.com; 3Scientific Support and Investigation Unit, CEVA-Phylaxia, 1107 Budapest, Hungary; tamas.mato@ceva.com; 4Department of Infectious Diseases and Public Health, Jockey Club College of Veterinary Medicine and Life Sciences, City University of Hong Kong, Hong Kong SAR, China

**Keywords:** Newcastle disease, paramyxovirus, broilers, Nepal, genotype VII.2

## Abstract

**Simple Summary:**

This study investigated the outbreaks of Newcastle disease in 10 broiler farms in Nepal from July to December 2021. The outbreaks occurred in farms that were either non-vaccinated or even fully vaccinated, indicating that the vaccination status of birds was not effective in preventing the disease. The initial disease diagnosis was performed based on typical signs of Newcastle disease such as limb paralysis, greenish diarrhea, torticollis, as well as pathological lesions including multifocal hemorrhages in the proventriculus, erosions and ulcers in the small intestine, and congestion and hemorrhages in the trachea. The molecular diagnosis aided the confirmation of the Newcastle disease virus and its genetic type. The presence of the avian influenza virus was ruled out with laboratory testing. The study highlights the occurrence of Newcastle disease outbreaks despite the fact that the suggested vaccination protocols were followed in the broiler flocks, emphasizing the need for comprehensive investigations into vaccines and genetic analysis.

**Abstract:**

Newcastle disease (ND) is a highly contagious viral disease caused by the paramyxovirus, which is a single-stranded ribonucleic acid (RNA) virus. This study was conducted to investigate ND outbreaks in 10 vaccinated or non-vaccinated broiler farms, collectively housing 9840 birds of various ages in the Chitwan and Nawalpur districts of Nepal from July to December 2021. Clinically, the affected birds exhibited symptoms such as limb paralysis, greenish diarrhea (seven out of ten flocks), torticollis (two out of ten flocks), inappetence, and drowsiness (ten out of ten flocks). Birds that succumbed during the clinical course underwent a necropsy for gross pathology and samples were collected for the histopathology and molecular diagnosis. The gross and microscopic examination revealed hemorrhages in the proventriculus, erosions and ulcers in the small intestine, congestion, as well as sero-mucosal hemorrhages in the trachea of affected birds, which are typical of ND. Rapid test kits further confirmed the presence of the ND virus antigen while excluding the avian influenza virus. Furthermore, M gene-based real time polymerase chain reaction (RT-PCR) was performed in the pooled samples from the affected birds and the presence of a velogenic strain of the ND virus was identified. The phylogenetic analysis of the RT-PCR positive strain based on the partial F gene nucleotide sequence revealed these strains as genotype VII.2 (formerly VIIi). The findings highlight the occurrence of clinical ND outbreaks in farms despite adherence to recommended vaccination protocols in broiler flocks, underscoring the need for a regular comprehensive investigation involving in-depth examinations of available vaccines and genetic analyses.

## 1. Introduction

Newcastle disease (ND) is a highly contagious viral disease caused by the avian paramyxovirus (APMV) belonging to the family Paramyxoviridae and the genus Avulavirus [1]. Altogether, there are 20 serogroups of avian paramyxoviruses, denoted as APMV-1 to APMV-20. Among them, APMV-1, with the virulent Newcastle disease virus (NDV), remains the most significant pathogen for poultry, although APMV-2, APMV-3, APMV-6, and APMV-7 are also known to cause diseases [2]. This virus contains a single-stranded, negative sense ribonucleic acid (RNA) and has helical capsid symmetry. There are altogether four pathotypes of the disease in chickens including velogenic, mesogenic, lentogenic, and asymptomatic enteric forms, but the groupings of these pathotypes may not always be clear-cut [3]. Velogenic forms are further divided into viscerotropic, velogenic, and neurotropic velogenic, based on the lesions induced in the visceral organs and nervous system.

Following natural exposure, the incubation period for the NDV has been observed to vary between 2 and 15 days with an average of 5–6 days [4]. The clinical symptoms exhibited by birds depend on factors such as the strain/pathotype, age, co-infection with other organisms, environmental stressors, and immunological status of the host [2]. Reduced feed and water intake followed by copious greenish diarrhea are common manifestations of the digestive form of ND [5]. Respiratory symptoms involve coughing, sneezing, gasping, and nasal discharge [6]. Additionally, birds may develop tracheal rales, which can be heard upon auscultation of the upper respiratory tract [7]. In severe cases, birds may exhibit facial swelling, conjunctivitis, and sinusitis [8]. Neurological signs are common in neurotropic ND, which include tremors, ataxia, torticollis, circling, incoordination, and paralysis. Although nervous signs are rare in viscerotropic velogenic form of disease, birds surviving the acute phase may develop similar signs [6]. The clinical manifestations of ND closely resemble those of influenza A, also known as the bird flu, complicating the diagnosis and intervention of the disease [9].

Microscopically, the gastrointestinal tract may display hemorrhagic foci associated with necrosis in the intestinal wall, especially in cases involving the viscerotropic virus [10]. As the disease progresses, button ulcers characterized by thickened, edematous, and hemorrhagic lesions begin to form at numerous sites of the gut-associated lymphoid tissue (GALT). The desquamation of the proventricular epithelial surface and gland may occur, while the intestine may exhibit the desquamation and necrosis of epithelial mucosal cells, characterized by the loss of villi and hemorrhaging and inflammatory cell infiltration in the submucosa [11]. Similarly, the proventriculus, exhibiting hemorrhages ranging from petechiae (particularly, at the tip of the gland) to ecchymosis, indicates viscerotropic velogenic ND [12]. Respiratory lesions are manifested as mucosal hemorrhaging, tracheal congestion, and air-sacculitis [13]. The mucosa of the trachea might be congested and edematic with an infiltration of lymphocytes and macrophages as well as a loss of cilia.

Despite the economic significance of ND in poultry, there exists a lack of comprehensive and up-to-date information about the incidence, distribution, and impact of the disease in chicken populations in Nepal. Additionally, the disease diagnosis in the field is largely based only on clinical symptoms and pathological lesions, which does not provide the molecular traits of the prevailing strains. Accurate data on the prevalence and pathologic characteristics of the NDV are crucial for the development and improvement of effective control and prevention strategies, and consequently mitigating the economic losses associated with the disease. Therefore, this study aimed to conduct a thorough assessment of ND outbreaks in the selected poultry farms located in the poultry-rich central region of Nepal with a special focus on the clinical, pathological, and molecular aspects of the disease. Utilizing a gross and microscopic examination, as well as molecular biology tools, the study intends to provide valuable insights into the occurrence, clinical presentations, progression of lesions, and strain of the NDV in clinical cases of ND within the country. 

## 2. Materials and Methods

### 2.1. Outbreak Summary

The study included a total of 10 broiler farms, collectively housing 9840 birds (Table 1). Clinical symptoms appeared in birds as early as 16 days of age, and as late as 36 days, with a mean age of 26.5 days. In two farms (Sukranagar and Bakulahar), no vaccinations were applied against ND. In another 5 farms (1 in Khdrauli, 2 in Tandi, 1 in Daldale, and 1 in Harkapur), a single F1 vaccination was administered at the age of 5 to 7 days. In remaining farms, F1 was applied at the age of 5 to 7 days, followed by a booster with the LaSota strain at 25 to 28 days. Both the F1 and LaSota vaccines used were live vaccines. F1 was administered either intraocularly or via drinking water while LaSota was given through drinking water only. In the prevailing practice, the administration of two vaccines is considered a full course of vaccination. The vaccine storage facilities were good in all the farms.

### 2.2. Clinical Observations and Rapid Screening of NDV

Clinical symptoms were recorded during farm visits. Flocks suspected of ND from clinical findings were subjected to screening for both NDV and AIV using rapid antigen test kits following the manufacturer’s recommendation (Bionote, Hwaseong, Republic of Korea). Among the estimated number of sick birds, which was 1935, 193 rapid tests were performed for NDV. From each bird, tracheal swabs were collected separately and pooled. Similarly, 20 rapid tests were performed for AIV (2 for each flock by sample pooling).

### 2.3. Gross Pathology

In total, 394 dead birds were necropsied following a standard procedure, which was 4% of the total birds. Briefly, the dead birds were dissected using sterile instruments, ensuring minimal damage to the internal organs. The body cavity was opened, and the appearance, location, and extent of any gross lesions were recorded using standardized pathology terminology. 

### 2.4. Histopathology

Twenty samples of each affected organs, including the proventriculus, intestine, and trachea from birds representing every flock (two birds/flock), were collected and preserved in 10% buffered formalin. The fixed tissue samples were processed to obtain thin sections, which were then mounted on glass slides, stained with Hematoxylin and Eosin, and observed under a microscope at various magnifications.

### 2.5. Molecular Analysis 

In order to detect the presence of the NDV gene and investigate the circulating strain of the NDV, four samples from the proventriculus of birds, representing flocks from Sukranagar and Bakulahar that were not vaccinated, from Khadrauli, which received one F1 vaccination, and from Jhuwani, which were vaccinated twice with F1 and Lasota, were collected on FTA cards and dispatched to the Himedia, Kolkata for molecular analysis. All birds exhibited prominent lesions suggestive of ND. After receiving samples, the elution of RNA from the FTA card was performed using an RNA rapid extraction solution (Invitrogen, Budapest, Hungary). The supernatants were collected and viral RNAs were extracted using the MagMAX Viral RNA Isolation Kit (Applied Biosystems, Carlsbad, CA, USA) according to the manufacturer’s instructions. Screening for the presence of NDV in the samples was performed with the M gene-based RT-real-time PCR method using TaqMan Fast Virus 1-step Master Mix (Applied Biosystems) and AMPV-1 Matrix primers and probe, which were previously described by Wise et al. [14].

The phylogenetic analysis of the RT-real-time PCR-positive strain was based on the partial F gene nucleotide sequence [15]. PCR products were purified using the QIAquick PCR purification kit (Qiagen, Hilden, Germany). Subsequently, purified PCR products were sequenced using the BigDye Terminator Cycle sequencing kit (Applied Biosystems) according to the manufacturer’s instructions. Sequences of the PCR products were determined using the dye terminator sequencing method and analyzed using the ABI PRISM^®^ 310 autosequencer (Applied Biosystems). The closest related sequence was search on the GenBank database using BLAST search. The phylogenetic analysis was performed using Molecular Evolutionary Genetics Analysis software (MEGA) version 7 [16]. Reference sequences were obtained from the GeneBank database based on the published information by Dimitrov et al. [17]. The comparison of nucleotide sequences was performed with Kimura’s two-parameters method, and the phylogenetic tree was constructed using the neighbor-joining method with a bootstrap test (1000 replicates). 

## 3. Results

### 3.1. Clinical Findings 

All the sick birds exhibited consistent clinical signs, including loss of appetite (in ten of ten flocks), resulting in reduced feed consumption or complete refusal to eat, followed by the onset of greenish diarrhea (in seven of ten flocks) within 24 to 48 h (Figure 1A). Notably, affected birds exhibited respiratory distress (in seven of ten flocks), characterized by difficulties in breathing, including labored breath, accelerated breathing rate, and, in severe cases, open-mouthed breathing or gasping for air. Additionally, some birds in five flocks showed signs of coughing, sneezing, or nasal discharge. Further manifestations noted during the examination of birds in all the flocks were fever, lethargy, weakness, and hunched posture, with ruffled feathers and reluctance to move. In two of ten flocks, birds showed neurological signs such as head twisting (torticollis) (Figure 1B) and paralysis (Figure 1C). 

### 3.2. Rapid Screening Test 

Among 193 tests, 180 were positive (93.27%) for the NDV (Table 1). However, all the tests performed for the AIV were found negative.

### 3.3. Necropsy Findings 

One of the most noteworthy and consistent findings was the presence of moderate to severe hemorrhaging in the proventriculus, the glandular part of the stomach. The hemorrhages ranged from petechiae (small red spots) to ecchymosis (larger areas of irregular hemorrhage) with petechiae being the most frequently observed manifestation (Figure 2A). 

Hemorrhagic ulcers, characterized by necrotic masses and mucosal ulcerations leading to the formation of granuloma (Button ulcers), was observed in the intestine (Figure 2B). However, the lesion was not as consistent as the proventricular hemorrhage. The elongated lesion sporadically appeared in the small intestine, which was visible from the serosal surface. 

Tracheal congestion and hemorrhages were prominently identified, leading to respiratory distress in the birds. These lesions were mainly localized in the upper trachea (cervical portion) (Figure 2C). The severity of the lesions showed a direct correlation with the level of respiratory discomfort experienced by the birds.

### 3.4. Histopathology Findings 

The villi in the proventriculus showed a reduction in size or degeneration, indicating atrophy (Figure 3A). The central and secondary ducts exhibited massive necrosis, hemorrhages, and alterations in the brush border. There was a presence of dilated and necrosed glandular ducts in the proventriculus, indicating cellular death and damage. Fibrosis, the formation of excess fibrous connective tissue, was noted at the junction of ducts within the proventriculus. Hemorrhaging was also observed in these areas. Likewise, the histopathological changes in the intestine were observed as villous necrosis. There was massive villous necrosis followed by the atrophy of villi (Figure 3B). These were the most consistent microscopic findings in the intestinal tissue samples. The necrosis was further accompanied with cellular infiltration within the crypts, indicating the intense scale of inflammation corresponding to the lesions of ND. The trachea had major lesions concentrated at the respiratory epithelium, which included ciliary necrosis followed by a loss of cilia (Figure 3C).

### 3.5. Molecular Findings 

All samples sent for a molecular biological analysis were positive for the NDV using RT real-time PCR testing, but only one sample contained enough NDV nucleic acid for further genetic characterization. The nucleotide sequence of this strain (chicken/Nepal/D5728-2-3/2021) is deposited in the GenBank under the accession number: PQ141584. Based on the phylogenetic analysis, the strain was grouped together with genotype VII.2 (former VIIi) velogenic NDV strains. The closest related sequence in the GenBank database was NDV/chicken/RPR/01/23 from India [18], showing 99.20% nucleotide sequence homology, which also groups together with genotype VII.2 strains (Figure 4). The multibasic amino acid sequence of the cleavage site was characteristic of velogenic strains (RRQKR’F), both in the case of the tested strain (chicken/Nepal/D5728-2-3/2021) and the closest sequence deposited in Genbank (NDV/chicken/RPR/01/23).

## 4. Discussion

ND is indeed one of the prominent poultry diseases worldwide that has significant consequences for the poultry industry, leading to high morbidity and mortality, decline in production, and ultimately resulting in substantial economic losses [2]. The high mortality associated with ND infection in the poultry industry can be attributed to its impact on several important organs such as the brain, lung, digestive tract, and lymphoid system [19]. Within the lymphoid system, the ND virus destroys lymphocytes and hampers antibody formation, resulting in the reduced efficacy of vaccination [20]. Most of the documented clinical presentations, gross and microscopic pathological lesions, such as intestinal and tracheal bleeding, pneumonia, intestinal necrosis, glandular necrosis, hemorrhaging, and lymphocyte destruction, were observed in this study. In addition, by utilizing qPCR techniques, this study determined the velogenic strain of ND as a cause of illness in the selected flocks. Thus, this study confirmed the outbreak as ND based on the consistent clinical manifestation, presentation of macro and micro lesions on the targeted tissues, and the subsequent molecular analysis in the vaccinated as well as non-vaccinated broiler flocks.

The ND outbreaks were observed in birds as early as 16 days and as late as 36 days old. The age at which a disease outbreak occurs can vary, depending on the specific disease and its characteristics. In the case of ND, the age of outbreak can be influenced by various factors such as the mode of transmission, immune status of the host, level of maternal immunity, and environmental conditions [21]. In some cases, ND outbreaks may occur in birds as young as two weeks of age. This can be attributed to the fact that young birds, especially chicks, generally have a less developed immune system. This may also be due to the low level of maternal-derived antibodies. Additionally, factors such as stress, overcrowding, and poor biosecurity practices at the early stages of life can increase the vulnerability of young birds to diseases like ND. Likewise, several immuno-suppressive diseases such as reovirus infection, infectious bursal disease, chicken infectious anemia, and mycotoxins may predispose the young chicks for ND infection. However, it is important to note that ND outbreaks can also occur later, around 3–4 weeks of age or even beyond. This can be due to factors such as the incubation period of the virus, exposure to the virus at a later stage, or the presence of other contributing factors such as concurrent infections or environmental stressors [22]. An outbreak later than 36 days in the broiler flocks was not observed, which might be because of the marketable age of birds.

In this study, it is important to note that the ND outbreaks were observed in eight vaccinated flocks and two non-vaccinated flocks. It was noted that in some flocks (five flocks), a partial vaccination course was administered, where they received the first dose but did not receive the second dose of the vaccine. Additionally, in three flocks, NDV infections occurred even after the completion of the full course of vaccination. There are multiple reasons why vaccinated birds can still become infected with the NDV. According to Wiedermann et al., these reasons can be broadly categorized into vaccine-related factors and host-related factors [23]. Vaccine-related factors include issues like the improper attenuation, storage, and administration of the vaccine. On the other hand, host-related factors involve genetic factors, age, health, and immune status. Factors contributing to vaccine interference include pre-existing immunity, the genetic and environmental background of the birds, vaccine schedule, and mode of delivery [24]. The occurrence of ND outbreaks in vaccinated flocks could be attributed to a combination of factors. One possible reason is the exposure of the vaccinated birds to high viral loads of the NDV [25]. This exposure can overwhelm the immunity provided by the vaccine, leading to the disease conditions. In addition, poor biosecurity practices in poultry farms can contribute to the outbreak. Inadequate biosecurity measures may allow the entry of the virus into the flock, despite their vaccination status [26]. Proper biosecurity protocols, such as controlling visitor access, disinfection procedures, and preventing contact with infected birds or contaminated materials, are crucial in preventing disease transmission [27]. Improper vaccine storage and handling can also play a role in the failure of vaccination [28]. Vaccines require specific storage conditions, including temperature control and protection from light. If vaccines are not stored and handled properly, their efficacy can be compromised, reducing the protection they provide to the vaccinated birds [29]. This study agrees with the opinion that a combination of factors, including exposure to high viral loads, poor biosecurity practices, and improper vaccine storage and handling, can contribute to ND outbreaks in vaccinated flocks. It is crucial to implement strict biosecurity measures and ensure proper vaccine storage and handling to enhance the effectiveness of vaccination programs. By addressing these factors, the risk of ND outbreaks can be minimized for the better protection of the health of vaccinated flocks. 

The observed clinical signs of the ND-affected birds included a mixture of digestive, respiratory, and neurological manifestations. Loss of appetite leading to partial or complete refusal of feed was consistently observed. Additionally, a decrease in feed and water intake followed by copious greenish diarrhea was noted, which could be attributed to damage caused by the viscerotropic velogenic ND (vvND) virus on the digestive tract of the birds [5]. The severity of greenish diarrhea varied, which could be influenced by the immune status of the birds and concurrent environmental stresses. Furthermore, the variation in severity might also be associated with the different routes of viral entry. Generally, the fecal–oral transmission of viruses tends to produce more pronounced digestive signs, while aerosol infection leads to marked respiratory problems [30]. Open mouth breathing observed in the affected birds could be due to viral multiplication in the upper respiratory tract or because of the general febrile condition [31]. Notably, nervous signs such as torticollis and lameness were rare, which are typically common manifestations of neurotropic ND [32]. Birds that survived the acute phase of vvND may develop neurological signs such as lameness and torticollis [33]. 

The study employed a rapid test antigen kit to detect the NDV antigen in suspicious cases. The descriptive statistical analysis of the test results revealed 93.26 percent of positive cases and 6.74 percent of negative cases for ND. The kit can be used as a rapid diagnostic tool for the detection of ND virus infection in the poultry industry. The use of test kits in screening ND (presumably referring to flocks of birds affected by ND) was found to be quite useful in terms of its precision and convenience. This implies that the test kits were able to accurately identify cases of ND in the flocks, and they were also easy to use in practical field conditions. This information highlights the practical benefits of using these test kits for screening and managing ND in flocks of birds in resource-limited settings like in Nepal. 

In the study, clinical findings were similar to those previously reported in experimental infection studies [10,34]. The fact that these findings align with the results from previous studies reinforces the consistency and validity of the observed lesions in the current study. The consistent observations of hemorrhaging in the proventriculus among the sick birds suggest a strong association between the presence of ND and the specific pathological manifestation. Similarly, the observed gross lesions in the intestine were found to resemble a similar study [10]. These lesions can be characterized in terms of hemorrhaging accompanied by necrosis and edema, which gives the appearance of button-shaped ulcers in the gut-associated lymphoid tissue (GALT). These lesions can be visible even from the serosal surface of the intestine, and their size and number tend to increase as the disease progresses. These findings are also consistent with previous observations [34]. The other literature suggested that the presence of hemorrhagic lesions accompanied by necrosis and edema, such as button ulcers in the intestine, is indicative of viscerotropic velogenic ND [35]. Overall, the consistency of these observed lesions in the intestine with previous research reinforces the association between the presence of viscerotropic velogenic ND. 

In addition, marked congestion and hemorrhages, exudation, and cellular infiltration with necrosis of the brush border were observed in the trachea of the affected birds. These findings suggest that the observed lesions in the trachea might have contributed to respiratory distress in the birds. The lesions were predominantly localized in the cervical or upper trachea. The tracheal lesion was preceded by the swelling of the ciliated columnar and mucous glands, epithelial cell proliferation, fibrinopurulent exudation, and lymphocytic infiltration. The severity of epithelial cell proliferation was particularly evident and severe in a previous study [36], suggesting consistency in the observed phenomenon. These observations collectively indicate the presence of significant pathological changes in the trachea, which likely contribute to the respiratory distress experienced by the birds. The consistency of these findings with previous studies strengthens the understanding of the disease and its impact on the respiratory system of affected birds.

In the present study, several histopathological changes were observed in the proventriculus, intestine, and respiratory epithelium of the affected birds. Hemorrhages in the proventriculus, the atrophy of villi and necrosis, the dilatation and necrosis of the duct system with fibrosis at the junction, and the hemorrhages at the brush border of the tertiary duct were evident. Brar et al. reported similar histopathological changes in the proventriculus of chickens infected with the ND virus [37]. These changes included glandular atrophy, epithelial necrosis, and mucosal inflammation. Furthermore, another report documented the shortening of proventricular papillae and lymphocytic infiltration in birds seven days post-infection [38], which is in agreement with the findings of the present study. In the intestine, massive villous necrosis, the infiltration of inflammatory cells in crypts, and the hemorrhaging of the surface epithelium were consistently observed. Villous atrophy, hemorrhages, and cellular infiltration were also seen. These observations correspond to the earlier findings [38,39]. Regarding the respiratory epithelium, consistent observations such as hemorrhages appeared at the junction of respiratory epithelium and cartilage along with ciliary necrosis (loss of cilia). These findings also align with an earlier study [39], the authors of which reported tracheal lesions characterized by hemorrhages and a loss of cilia. Similarly, the mild to moderate infiltration of lymphocytic cells, moderate degeneration of the epithelial cells, and infiltration of mononuclear inflammatory cells, usually in submucosal areas and hemorrhagic foci accompanied by necrosis in the mucosal lymphoid tissue, were similar to the findings of Gopal et al. [40]. Overall, the observed histopathological changes in the proventriculus, intestine, and respiratory epithelium are consistent with previous studies and provide insight into the impact of the disease on these organs and tissues, confirming the presence of the NDV. 

In regard to the molecular characterization of suspected ND cases, the real time-PCR followed by phylogenetic analysis revealed the velogenic strain (genotype VII.2) in the samples processed through molecular techniques. The findings align with the findings of Napit et al. [41], who investigated the outbreaks of ND from Kathmandu and found out that genotype VII was the cause of the outbreak. The same authors also reported nonvirulent genotype II and I to be circulating in the country. 

## 5. Conclusions

The study concluded by documenting ND outbreaks in the Central region of Nepal, specifically in the Chitwan and Nawalpur districts, from July to December 2021 in vaccinated and non-vaccinated broiler flocks. The clinical presentations, rapid test antigen test, necropsy, gross and microscopic pathology, along with the modern molecular technique (qPCR) and phylogenetic analysis indicated that the vvNDV was responsible for the outbreaks. The severity of the disease was evident as all farms, which experienced 100% mortality rates regardless of the vaccination status. The phylogenetic analyses suggest the presence of genotype VII.2. This study confirms the capacity of sub-genotype VII.2 viruses to cause outbreaks in vaccinated birds and the mechanism for this remains unknown and need to be investigated. The generated data provide a comprehensive characterization and further knowledge on viruses circulating in Nepal and will facilitate future studies of the genetic diversity of NDV.

## Figures and Tables

**Figure 1 animals-14-02423-f001:**
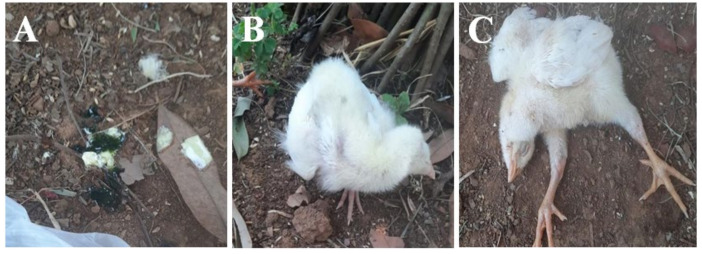
Clinical findings in ND infected birds. Watery greenish diarrhea (**A**); torticollis (**B**); bilateral limb paralysis (**C**).

**Figure 2 animals-14-02423-f002:**
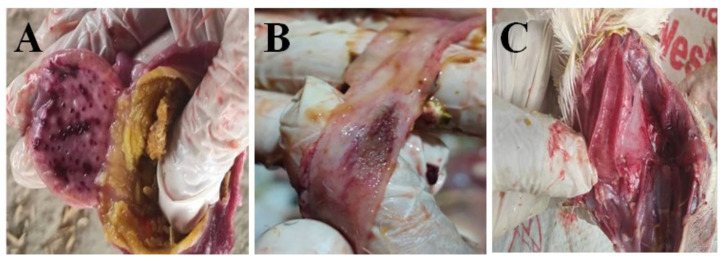
Macroscopic lesions in broiler infected with ND. Petechial hemorrhage in proventriculus (**A**); button ulcer in intestine (**B**); hemorrhage in upper trachea (**C**).

**Figure 3 animals-14-02423-f003:**
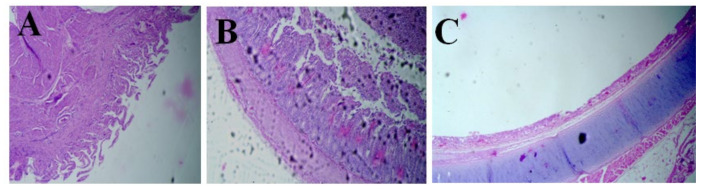
Microscopic lesions in broiler infected with ND. Atrophy of proventricular villi (**A**), intestinal villous necrosis (**B**), loss of cilia in trachea (**C**).

**Figure 4 animals-14-02423-f004:**
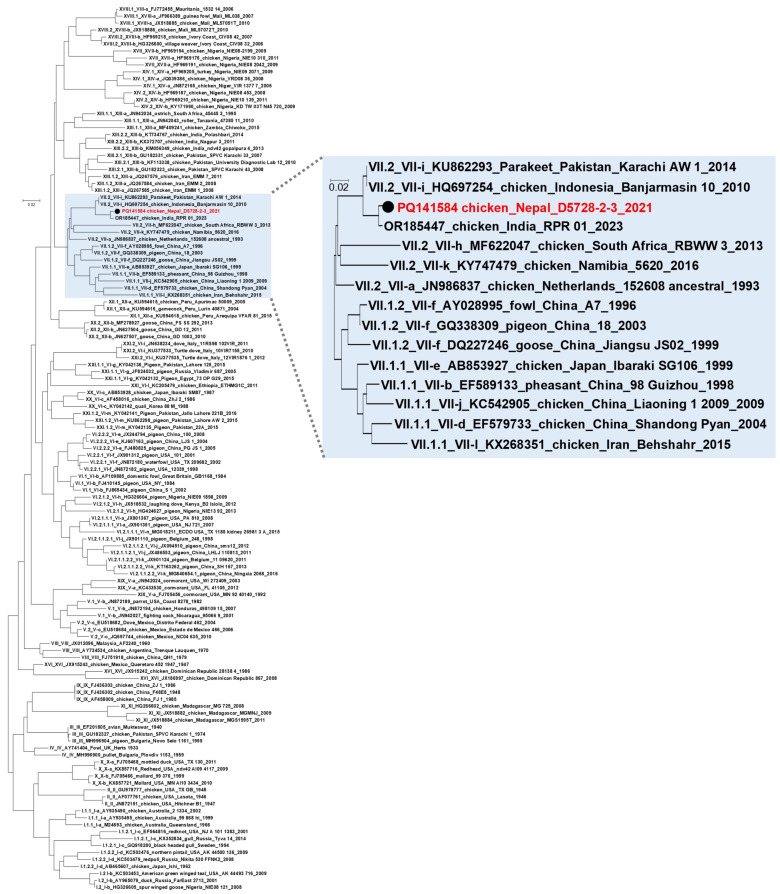
Phylogenetic tree generated by using nucleotide sequences of the partial F gene. The NDV strain tested in this study is indicated with a black dot and red letters. Other sequences included in the tree to obtain proper typing were obtained from GeneBank database based on the published information by Dimitrov et al. [17].

**Table 1 animals-14-02423-t001:** Summary of the studied outbreaks of ND in Chitwan and Nawalpur and details of ND screening with rapid test kits.

District	Farm Location	Flock Size	Age of Onset (Days)	Vaccine Administered ^1^	Vaccinated on (days)	Eventual Fatality	Estimated Number of Sick Birds at Reporting	Positive/Tested Samples (ND Rapid Test Kits)
Chitwan	Khadrauli	500	19	F1	6	100%	125	11/12
	Shukranagar	790	22	-	-	100%	160	14/16
	Jhuwani,	600	30	F1	7	100%	100	9/10
				LaSota	28			
	Tandi	400	30	F1	7	100%	150	15/15
	Tandi	1000	16	F1	5	100%	200	18/20
	Bakulahar	500	36	-	-	100%	250	25/25
	Parbatipur	650	34	F1	7	100%	150	14/15
				LaSota	28			
Nawalpur	Daldale	3200	19	F1	6	100%	400	37/40
	Harkapur	700	32	F1	6	100%	200	19/20
	Gaidakot	1500	27	F1	6	100%	200	18/20
				LaSota	25			
	Total	9840	$\overset{-}{x}$ = 26.5				1935	180/193

^1^ LaSota vaccine was administered as a booster.

## Data Availability

All the data are provided in the manuscript.

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
