# Peer review of "Clinicopathological and Molecular Investigation of Newcastle Disease Outbreaks in Vaccinated and Non-Vaccinated Broiler Chicken Flocks in Nepal"

_animals, 2024, doi:10.3390/ani14162423_

Round 1

Reviewer 1 Report

Comments and Suggestions for Authors

The manuscript [animals-3126274], entitled “Clinicopathological And Molecular Investigation Of Newcastle 2 Disease Outbreaks In Vaccinated and Non-vaccinated Broiler 3 Chicken Flocks In Nepal” by Prof. Surya Paudel et al., reports findings of the outbreaks of Newcastle disease in 10 broiler farms in 15 Nepal from July to December 2021.

This is an interesting and important study.

Some concerns are listed below for consideration in revision.

      Major concerns:

1.      Section 2 Gross pathology, suggested to improve the description (lines 119-120), I have not understood the protocol.

2.      Section 2 Molecular analysis, suggested to improve the description (lines 139-140), I have not understood the protocol for real PCR.

3.      Section 3 Clinical findings, suggested to add more information about the clinical finding, such as the appearance rate for each clinical signs in the chickens.

4.      Section 3 Molecular findings, only one sequence for the phylogenetic analysis is limited, strains from the ten flocks need sequencing, and the obtained Partial F gene sequence should be deposited into the GENBANK and got the accession NO.s.

Some specific minor concerns:

5.      Abstract: the positive rate and appearance rate for each clinical signs should be added.

6.      Section 2 Molecular analysis, suggested to add the suppliers for the Mega software, the bootstrap protocols for phylogenetic analysis should also be added.

Author Response

Major concerns:

  1. Section 2 Gross pathology, suggested to improve the description (lines 119-120), I have not understood the protocol.

Authors' response: We greatly appreciate the time and effort made by the reviewer in reviewing our manuscript.

Please note that line numbers provided in the authors' response below correspond to the revised manuscript in tract change. 

The description of postmortem procedure is improved as suggested. Line 128-132.

  1. Section 2 Molecular analysis, suggested to improve the description (lines 139-140), I have not understood the protocol for real PCR.

Authors' response: The description is improved as suggested. Accordingly, an additional citation is added to clarify the phylogenetic analysis. Line 146-159.

  1. Section 3 Clinical findings, suggested to add more information about the clinical finding, such as the appearance rate for each clinical signs in the chickens.

Authors' response: Details regarding the appearance rates are provided as suggested. Line 177-187.

  1. Section 3 Molecular findings, only one sequence for the phylogenetic analysis is limited, strains from the ten flocks need sequencing, and the obtained Partial F gene sequence should be deposited into the GENBANK and got the accession NO.s.

Authors' response: Out of four samples obtained, only one sample was found suitable for sequence analysis. This information is now mentioned in the manuscript. Partial F gene sequence is also deposited in Genbank and the accession number is provided in the manuscript. Line 302-312.

Some specific minor concerns:

  1. Abstract: the positive rate and appearance rate for each clinical signs should be added.

Authors' response: The information is provided as suggested. Line 30-31.

  1. Section 2 Molecular analysis, suggested to add the suppliers for the Mega software, the bootstrap protocols for phylogenetic analysis should also be added.

Authors' response:Mega software is a freely available, which can be downloaded from the internet (https://www.megasoftware.net/). Reference is now added in the manuscript. Line 168-169.

Reviewer 2 Report

Comments and Suggestions for Authors

The following points should be considered for improvement  

The title should be informative, concise and accurately reflect the contents of the study. 

The data provided in this paper is too small for a complete full length research article. 

The objectives should be clearly described and let the readers know why this study is unique in nature and contents. 

The figure 1 provided does not provide any scientific information regarding this study. Provide more potential images of the gross and microscopic findings.

the discussion is weak, the authors should provide detailed discussion. Moreover, the abbreviations should be carefully used. 

Author Response

The following points should be considered for improvement  

The title should be informative, concise and accurately reflect the contents of the study.

Authors' response: We greatly appreciate the time and effort made by the reviewer in reviewing our manuscript.

Please note that line numbers provided in the authors' response below correspond to the revised manuscript in tract change. 

In our opinion, the title accurately reflects the work done in this study. Therefore, we prefer to keep the original title as it is.  

The data provided in this paper is too small for a complete full length research article. 

Authors' response: We thank the reviewer for the comment. We do understand that molecular biology data may be too small. However, we are afraid that other clinical and pathological data might have been overlooked. Importantly, the study shows serious outbreaks of Newcastle disease in vaccinated as well as in non-vaccinated flocks with similar extent of severity. The information is foreseen to be crucial for the management of the disease. 

The objectives should be clearly described and let the readers know why this study is unique in nature and contents. 

Authors' response: The objective is revised as suggested. Line 85-98.

The figure 1 provided does not provide any scientific information regarding this study. Provide more potential images of the gross and microscopic findings.

Authors' response: Figure 1 shows greenish diarrhoea, torticollis and limp paralysis, which is commonly used by field veterinarians for initial diagnosis of Newcastle disease. So, we believe that the illustrations are relevant in the manuscript. Gross and microscopic findinsg are provided in Figures 2 and 3, respectively.

the discussion is weak, the authors should provide detailed discussion. Moreover, the abbreviations should be carefully used. 

Authors' response: The discussion is revised as suggested. Abbreviations are carefully checked and revised.

Reviewer 3 Report

Comments and Suggestions for Authors

·         Why were only 20 birds evaluated by histopathology out of the 394 birds necropsied?

·         Please include information of why additional molecular sequence data was not obtained. Only 4 samples tested by PCR and sequencing.

·         Indicate which flocks were evaluated by PCR and sequencing.

·         More data would be beneficial for determining genomic/phylogenetic relatedness. The value of the study would be greatly increased by determining whether the virus infecting each flock was the same strain. This would require whole genome sequence data which was not attempted. All genetic conclusions were based on a short segment of the fusion gene. Were all 4 molecular samples successfully sequenced? Please address this and the basis for limited molecular data.

·         More information would be beneficial about the vaccines themselves. Were they were inactivated or live? Which flocks received only a single vaccination and which received a full course? What signifies a full course, # days between vaccinations? How were the vaccines stored at each facility? How were they administered?

·         It would be beneficial to indicate Lasota was in fact a booster in Table 1. It is currently implied in the table and the reader must refer back to the Outbreak summary section to confirm.

·         Were there any investigations into the source of virus introduction within each facility?

·         Line 387-389 This sentence is redundant with the previous sentence.

·         Was NDV/chicken/RPR/01/23 from India also indicated to be viscerotropic velogenic?

Author Response

Why were only 20 birds evaluated by histopathology out of the 394 birds necropsied?

Authors' response: We greatly appreciate the time and effort made by the reviewer in reviewing our manuscript.

Please note that line numbers mentioned below in authors' response correspond to the revised manuscript in track change.

From each flock, 2 birds were selected and samples of proventriculus, intestine and trachea were processed for histopathology. Thus, in total 60 histopathological slides were prepared representing every flock included in the study and all the important organs relevant for the ND. Based on the nature of the disease characteristics, we do not expect any new information on the microscopic findings even if sample size was increased. Thus, in view of the limited resources, the sample size was determined to be 2 birds/flock with three organs from each.  

Please include information of why additional molecular sequence data was not obtained. Only 4 samples tested by PCR and sequencing.

Authors' response: The information is provided in the manuscript as suggested. Line 302-304.

Indicate which flocks were evaluated by PCR and sequencing.

Authors' response: The information is now provided in the manuscript. Line 146-151.

  • More data would be beneficial for determining genomic/phylogenetic relatedness. The value of the study would be greatly increased by determining whether the virus infecting each flock was the same strain. This would require whole genome sequence data which was not attempted. All genetic conclusions were based on a short segment of the fusion gene. Were all 4 molecular samples successfully sequenced? Please address this and the basis for limited molecular data.

Authors' response: Unfortunately, only one sample was suitable for sequencing. The short sequence obtained results in the same topology as the whole F gene-based analysis, meaning that it allocates the sequences into genotypes appropriately. The relatively small sequence analyzed contains the cleavages site, which is the main determinant of pathogenicity, therefore the obtained sequence is also suitable for prediction of the pathogenicity of the strain. Based on these, the sequence analysis performed was suitable to draw reliable conclusions on the genotype and virulence of the NDV strain.

Since the sample submission was on done in FTA cards, it is hard, if possible, at all, to obtain long sequences due to the fragmentation of the RNA strands and the lack of high quantity of viral RNA. Full genome (or full gene) sequencing approaches mostly use virus isolates, which is not possible from FTA cards.

  • More information would be beneficial about the vaccines themselves. Were they were inactivated or live? Which flocks received only a single vaccination and which received a full course? What signifies a full course, # days between vaccinations? How were the vaccines stored at each facility? How were they administered?

Authors' response:

The information is provided in the manuscript as suggested. Line 112-115.

  • It would be beneficial to indicate Lasota was in fact a booster in Table 1. It is currently implied in the table and the reader must refer back to the Outbreak summary section to confirm.

Authors' response: The information is now provided in Tabel 1 as suggested.

  • Were there any investigations into the source of virus introduction within each facility?

Authors' response: Identifying the sources of virus introduction was not the scope of this study therefore it was not done.

  • Line 387-389 This sentence is redundant with the previous sentence.

Authors' response: The sentence is rephrased.

  • Was NDV/chicken/RPR/01/23 from India also indicated to be viscerotropic velogenic?

Authors' response: In the scientific article describing the genetic analysis of this strain there is no enough metadata to support that the strain was viscerotropic velogenic. Based on the fact that it was isolated from chickens showing ND symptoms (which were not described in details) and the presence of the multibasic cleavage site, its should be velogenic. Ref: Reddy et al. Gene Reports 34 (2024) 101884.

Round 2

Reviewer 1 Report

Comments and Suggestions for Authors

I have no other concern.